# Metabolic Syndrome and Overactive Bladder Syndrome May Share Common Pathophysiologies

**DOI:** 10.3390/biomedicines10081957

**Published:** 2022-08-12

**Authors:** Lin-Nei Hsu, Ju-Chuan Hu, Po-Yen Chen, Wei-Chia Lee, Yao-Chi Chuang

**Affiliations:** 1Department of Urology, An Nan Hospital, China Medical University, Tainan City 833, Taiwan; 2Division of Urology, Department of Surgery, Taichung Veterans General Hospital, Taichung 407, Taiwan; 3Division of Urology, Yunlin Chang Gung Memorial Hospital, Chang Gung University College of Medicine, Yunlin 638, Taiwan; 4Division of Urology, Kaohsiung Chang Gung Memorial Hospital, Chang Gung University College of Medicine, Kaohsiung 807, Taiwan

**Keywords:** insulin resistance, metabolic syndrome, microbiota, neuropathy, obesity, overactive bladder, proinflammation

## Abstract

Metabolic syndrome (MetS) is defined by a group of cardiovascular risk factors, including impaired glucose tolerance, central obesity, hypertension, and dyslipidemia. Overactive bladder (OAB) syndrome consists of symptoms such as urinary urgency, frequency, and nocturia with or without urge incontinence. The high prevalences of metabolic syndrome (MetS) and overactive bladder (OAB) worldwide affect quality of life and cause profound negative impacts on the social economy. Accumulated evidence suggests that MetS might contribute to the underlying mechanisms for developing OAB, and MetS-associated OAB could be a subtype of OAB. However, how could these two syndromes interact with each other? Based on results of animal studies and observations in epidemiological studies, we summarized the common pathophysiologies existing between MetS and OAB, including autonomic and peripheral neuropathies, chronic ischemia, proinflammatory status, dysregulation of nutrient-sensing pathways (e.g., insulin resistance at the bladder mucosa and excessive succinate intake), and the probable role of dysbiosis. Since the MetS-associated OAB is a subtype of OAB with distinctive pathophysiologies, the regular and non-specific medications, such as antimuscarinics, beta-3 agonist, and botulinum toxin injection, might lead to unsatisfying results. Understanding the pathophysiologies of MetS-associated OAB might benefit future studies exploring novel biomarkers for diagnosis and therapeutic targets on both MetS and OAB.

## 1. Introduction

Metabolic syndrome (MetS) and overactive bladder (OAB) syndrome both have a high global prevalence (>25% and 16–23%, respectively) and affect public health. They also have a high economic cost in our societies [1,2,3,4]. At present, scientists consider that MetS and OAB might share common pathophysiologies [5]. In addition, MetS-associated OAB should be an important subtype of OAB syndrome [6]. A syndrome is defined as a group of symptoms or signs which usually occur together, or a condition represented by a set of related symptoms. In 1988, Reaven originated the concept of “syndrome X”, which was later renamed “MetS” and identified by patients with a group of risk factors in coronary heart disease or type 2 diabetes [7]. Each MetS component is an independent risk factor for cardiovascular disease. Moreover, the combination of these factors elevates the severity of cardiovascular disease and increases the risk of type 2 diabetes. On the other hand, the term OAB was postulated as the title in a consensus conference, in 1999 [8]. Additionally, the International Continence Society (ICS) in 2002 stated, “Urgency, with or without urgent incontinence, usually with frequency and nocturia, can be described as ‘OAB syndrome’” [9]. In this review, we are trying to disclose the relationship between MetS and OAB syndrome: why these two seemingly unrelated syndromes (i.e., MetS and OAB) might have similar underlying pathophysiologies (Figure 1).

## 2. Pathophysiology of MetS

Insulin resistance and central obesity are the cardinal features of MetS. Many expert groups and international organizations define the MetS with different parameters (Table 1) [10,11,12,13,14,15]. In 1998, the World Health Organization (WHO) first put forward insulin resistance as being included in the pathophysiology of MetS and supported the role of prediabetic status in MetS [10]. Furthermore, the European Group for the Study of Insulin Resistance emphasizes the importance of hyperinsulinemia in MetS, but excludes the influence of microalbuminuria [11]. Then, the National Cholesterol Education Program (NCEP) Adult Treatment Panel III (ATP III) proposed a new summary using waist circumference (i.e., central obesity), dyslipidemia, hypertension, and fasting glucose for MetS [12,13]. In 2005, the MetS was more precisely defined by the International Diabetes Federation (IDF) [14]. The abdominal obesity was considered as a necessary diagnostic component and the simple tool in screening of MetS. Currently, the definitions of MetS provided by NCEP: ATP III and IDF are popularly applied worldwide. However, the new set of criteria with racial specific cut-offs should be taken into account for feasible and accurate diagnosis of individuals with MetS [15].

Insulin resistance can be considered as a defect in the ability of insulin to facilitate glucose disposal and poor response to insulin in tissues, such as skeletal muscle, liver, and white adipose tissue [16]. Once insulin resistance occurs, the pancreas will increase insulin to compensate, and fasting plasma insulin levels increase. The loss of coordination in glucose-lowering response involves the suppression of the endogenous glucose production suppression of lipolysis, cellular uptake of plasma glucose, and net glycogen synthesis, which will induce glucose intolerance, dyslipidemia, endothelial dysfunction, obesity, and inflammation [16]. The real-time feedback circuit linking insulin resistance and metabolic perturbations complicates the “chicken–egg” problem of identifying the primary pathogenesis.

In observing the over-nutrition status in developed nations, the rising prevalence of metabolic syndrome may be attributed to the excessive body fat and pathogenic adipose tissue, so-called “adiposopathy” [17]. In fact, adipose tissue could be an endocrine and active immune organ [17,18]. Adiposopathy is defined as pathogenic adipose tissue that is promoted by positive caloric balance and sedentary lifestyle in genetically and environmentally susceptible patients [17]. Anatomically, adiposopathy manifests adipocyte hypertrophy, visceral adipose tissue accumulation, adipose tissue growth that exceeds vascular supply, and ectopic fat deposition in patients. In pathophysiology, adiposopathy causes adverse metabolic and consequences resulting in metabolic perturbations. 

## 3. Pathophysiologies of OAB Associated with MetS

By the definition provided by ICS, OAB syndrome is an idiopathic bladder disorder, and its cardinal symptom is urinary urgency [9]. Therefore, the etiologies of OAB are unclear and sophisticated, from peripheral bladder dysfunction to the central sensitization of patients [19,20]. In addition, the lifestyles of patients may have a great impact on the development of OAB syndrome. For instance, caffeine intake may elicit an irritative bladder and slow-paced respiration may improve OAB syndrome among patients [21,22]. For the convenience of scientific investigation, the pathophysiologies of OAB could be classified as myogenic, neurogenic, urotheliogenic, and other specific conditions (MetS, type 2 diabetes, proinflammation, benign prostatic hyperplasia (BPH), affective disorders, urinary microbiota, etc.) [19].

In 2005, Rohrmann et al. first observed that men with MetS had an increased risk of nocturia, incomplete bladder emptying, weak stream, and hesitancy [23]. In an epidemiological investigation, Yu et al. demonstrated that hyperlipidemia is associated with presentations of OAB in Taiwanese women [24]. Sufficient evidence suggested that obesity could be an independent risk factor of OAB in female patients [25,26,27]. Recently, the CARDIA study corroborates that bladder health and cardiovascular health among women may share common factors, including lower body mass index and the absence of MetS [28].

From traditional viewpoints, increased nerve activates in vesical afferents and efferents, pelvic ischemia, increased oxidative stress, and chronic low-grade inflammation would be the candidates to link between MetS and OAB [19,29,30]. Recently, the nutrient-sensing pathway was found to be involved in the bladder dysfunction in rat models of MetS, including insulin resistance at bladder mucosa and a chronic increase in circulated succinate level [31,32]. Prospectively, dysbiosis in MetS might play a role in developing OAB. Due to distinctive pathophysiologies in MetS associated OAB, the treatment modalities of OAB should be reconsidered in the clinic [6].

## 4. Impaired Vesical Afferents and Efferents Activity

In investigating the urinary symptoms of patients with BPH, researchers found autonomic nervous system overactivity could link between MetS and lower urinary tract symptoms secondary to benign prostatic hyperplasia [33]. The use of an α_1_-adrenergic blocker, doxazosin, could improve glucose intolerance in patients under hyperinsulinemia [34] and also improve patients with bladder neck dysfunction [35]. In an animal study, Tong et al. reported that a high-fructose diet may induce traits of MetS (i.e., hypertension, hypertriglyceridemia, insulin resistance, and obesity) and bladder overactivity in male rats [36]. The upregulation of M_2_ and M_3_ muscarinic receptors in both bladder mucosa and detrusor suggested the alterations of vesical cholinergic system in rats. For the non-adrenergic and non-cholinergic system, Chung et al. observed abnormal pelvic nerve activity, along with impaired ATP-mediated contraction in cystometry as the ATP content of the bladder decreased in the 6-month fructose-fed male rat model [37]. Furthermore, Lee et al. found female fructose-fed rats may have bladder oversensitivity in cystometry during intravesical acidic ATP solution stimulation through vesical C-fiber transmission by upregulation of mucosal purinergic P2X_3_ receptor and TRPV1 receptor [38]. Hence, evidence suggested that traits of MetS might alter the transmission of autonomic efferents and peripheral sensory afferents of the bladder.

## 5. Chronic Bladder Ischemia

### 5.1. Pelvis Ischemia

One of the postulated etiologies of OAB is chronic bladder ischemia [39]. Since the MetS syndrome consists of a cluster of risk factors in cardiovascular diseases, atherosclerosis, through its related negative impact on bladder ischemia, is a potential mechanism for the development of OAB syndrome. In epidemiological studies, researchers reported that the severity of lower urinary tract symptoms is the risk factor of adverse cardiac events [40] and may increase the cardiovascular risk in terms of Framingham risk score [41], particularly in total cholesterol level. The vascular supply of the urinary bladder originates from the branches of the internal iliac artery, including superior, middle, and inferior vesical arteries. The bifurcation of iliac arteries is particularly vulnerable to atherosclerotic lesions. Atherosclerotic obstructive changes at the iliac bifurcation may affect the distal vasculature and bladder blood flow. In a series of studies [42,43,44], Azadzoi at al. used arterial balloon-induced endothelial injury combined with a 0.5% cholesterol diet to mimic chronic bladder ischemia caused by atherosclerosis. Bladder overactivity, along with bladder fibrosis, oxidative stress sensitive genes, and neurodegeneration, were observed in this rabbit model of chronic bladder ischemia. In a rat model of pelvis ischemia, Tai et al. reported that pelvis ischemia may enhance endoplasmic reticulum stress, autophagy, and apoptosis of the bladder [45]. Such kinds of bladder dysfunction could not be well managed by the mainstream OAB medication, such as antimuscarinics.

### 5.2. MetS Associated OAB Secondary to BPH

The MetS may further deteriorate bladder hypoperfusion secondary to BPH in patients [30]. Under circumstances of bladder outlet obstruction, the bladder will try to evacuate the urine by enhancing detrusor pressure, and consequently reduce the blood perfusion due to a high intravesical pressure. During micturition cycle, the cyclic ischemia and reperfusion occur. Koritsiadis et al. revealed the overexpression of hypoxia-inducible factor-1α could be observed in the bladders of patients with bladder outlet obstruction [46]. On the other hand, hyperinsulinemia and adiposopathy might facilitate the growth of prostate tissue [47]. Insulin-like growth factor 1 has been shown to promote prostate epithelial growth. Nandeesha et al. found that fasting hyperinsulinemia (plasma insulin level higher than 13 mU/mL) is a risk factor to increase prostate volume in patients [48]. Moreover, adiposopathy may result in dysregulation of adipokines, which may lead to development of BPH [47]. In the Baltimore Longitudinal Study of Aging, researchers reported that each kg/m2 increase in body mass index was associated with a 0.41 mL increase in prostate volume [49]. Thus, the interaction between MetS and BPH may further worsen the bladder blood perfusion and promote the progression of OAB syndrome.

## 6. Chronic Low Grade Proinflammatory State

In searching for the biomarkers of OAB, scientists found several inflammatory mediators in the sera or urine among OAB patients, such as C-reactive protein (CRP), nerve growth factor (NGF), brain-derived neurotrophic factor (BDNF), and prostaglandin E2 (PGE2) [19,50]. Interestingly, MetS per se is associated with a state of chronic low-grade inflammation, including elevated levels of inflammation markers, such as C-reactive protein (CRP), and proinflammatory cytokines, such as tumor necrosis factor-α, interleukin-6, and interleukin-8 [29]. Additionally, chronic inflammation has been proposed as a candidate mechanism at the crossroad between OAB and MetS. Therefore, we may hypothesize that metabolic inflammation results in tissue fibrosis, which progresses to inflammation initiation; aberrant wound healing, collagen deposition, and extracellular matrix remodeling; and increased stiffness of the bladder [29].

### 6.1. CRP

CRP releases from the liver into the bloodstream in response to the inflammatory process. Elevated serum CRP associated with OAB presentations supports the role of inflammation as an etiology. In the Boston Area Community Health study, Kupelian et al. reported that the prevalence of OAB increases with CRP levels in both genders [51]. Hsiao et al. [52] and Chung et al. [53] suggested that the OAB patients with urgent incontinence had higher serum CRP levels than patients without urgent incontinence. Meanwhile, Mirhafez et al. observed that high-sensitivity CRP could be a biomarker for MetS patients in a cohort study [54]. In that study, most features of MetS were associated with an increase in high-sensitivity serum CRP.

### 6.2. Neurotrophins

The role of neurotrophins, including NGF and BDNF, on the OAB biomarkers, has been emphasized [19,50]. Neurotrophins are growth factors required by neurons for survival, and also maintain a broad range of activities in the central and peripheral nervous system. In neurogenic OAB, it is thought of as a local shift away from Aδ afferent fiber involved in normal voiding to abnormal C fiber activity in pathologic state [50]. Through the neuroplasticity process, it may be triggered by neurotrophic factors affecting vesical afferents. Neurotrophins act via the p75 neurotrophin receptor and the family of tyrosine kinase receptors [55]. For MetS, the serum level of NGF is positively related to obesity and other inflammatory markers of people [56]. According to neurotrophic hypothesis, researchers can observe an increase NGF level in the early stages of MetS, following a decreased NGF level in the generalized stage in population [55]. Dagdeviren and Cengiz checked the serum NGF levels in OAB and MetS women and found the mean serum NGF levels of OAB and MetS/OAB women were significantly higher than those of normal controls [57].

### 6.3. Prostaglandins

Elevated urinary PGE2 or metabolites could be a biomarker for MetS and OAB patients [19,50,58]. PGs derived from fatty acid are mediators of numerous physiological effects by acting as local messengers for paracrine. PGE2 is generated by the metabolism of arachidonic acid by cyclooxygenase and PGE synthesis. It has four G protein-coupled receptors: EP1 through EP4. Yasui et al. suggested PGE2 via EP4 signaling could improve obesity-related adipose tissue inflammation and insulin resistance [59]. In a human study, Pawelzik and colleagues reported that urinary PGD2 and E2 metabolites were associated with obesity, dyslipidemia, and insulin resistance [58]. PGs are also locally synthesized in the bladder muscle and mucosa. Detrusor muscle stretch, bladder nerve stimulation, bladder mucosa damage, and inflammatory mediators can promote PGs synthesis [50]. In animals and human beings, PGE2 instilled into bladder can elicit uninhibited detrusor contraction [19].

## 7. Dysregulation of Nutrient-Sensing Pathways

Certainly, environmental factors have a great impact on the development of MetS and OAB. Fundamentally, nutrient intake and activating nutrient-sensing pathways are the convenient paths to communicating between the environment and the organism [60]. Scientists indicated two probable nutrient-sensing mechanisms: insulin resistance at the bladder mucosa and excessive succinate intake [31,32], which may impair bladder function directly under metabolic diseases.

### 7.1. Insulin Resistance at the Bladder Mucosa

In studying the relationship between MetS and OAB syndrome, Uzun et al. found out the association between OAB and insulin resistance in women [61]. They reported that female OAB patients have hyperinsulinemia and higher homeostasis model assessment of insulin resistance (HOMA-IR) values than controls. In an obese mice model, Leiria et al. demonstrated that defective insulin action in bladder mucosa impaired the detrusor relaxation and contributed to bladder overactivity [31]. Using phenylbutyric acid (a chemical chaperone) to inhibit endoplasmic reticulum stress activation of the unfolded protein response, they restored the PI3k/AKT/eNOS pathway of bladder mucosa in the obese mice, along with glucose homeostasis. Given that insulin resistance is a major feature of MetS, reduced insulin action in the bladder mucosa could be a common mechanism for eliciting OAB symptoms under circumstances, such as those of MetS. Thus, Lee et al. reported that either insulin resistance at the bladder mucosa obtained from epigenetic regulations by maternal fructose exposure or post-weaning fructose intake by rats may have impaired detrusor relaxation and bladder overactivity due to insufficient production of bladder NO/cGMP [62]. In another study, Lee et al. also demonstrated tadalafil (a PDE 5 inhibitor) can restore the canonical signaling pathway of insulin at the bladder mucosa and improve the relaxation of detrusor and bladder overactivity in fructose-fed rats [63].

### 7.2. Excessive Succinate Intake

Succinate is an essential intermediate of the tricarboxylic acid cycle that exerts pleiotropic roles through GPR91 (a G protein-coupled receptor), including worse hypertension, metabolic signaling, and impaired glucose intolerance [64,65]. Researchers reported that plasma succinate levels are linked to proinflammation and more visceral adipose tissue in young adults [66]. Mossa et al. reported that urothelial cells and detrusor muscle can express GPR91 [67]. Succinate via GPR91 can inhibit forskolin-stimulated cAMP production in urothelial cells. In addition, incubation of urothelial cells with succinate potently increased iNOS synthesis and secretion of nitric oxide, and decreased secretion of PGE2. Furthermore, Flores et al. found excessive succinate intake would impair bladder function and promote bladder fibrosis in rats [32]. In culturing urothelial cells and smooth muscle cells of rats, Mossa et al. demonstrated the counteraction between mirabegron (a β3 adrenergic receptor agonist) and succinate in the production of cAMP in smooth muscle cells [68]. Taken together, these studies suggested that elevated levels of succinate in circulation will deteriorate features of MetS and impair bladder function.

## 8. The Role of Dysbiosis between MetS and OAB Syndrome in Perspective

### 8.1. Urinary Microbiome Might Contribute to OAB Syndrome

At present, urine is no longer considered sterile or merely a waste product under the modern paradigm [69]. In fact, urine is considered to have chemical, physical, and biological effects on the homeostasis of bladder function. Urine per se could act as an irritant chemical and cause irritative symptoms without the protection of healthy urothelium, and the urinary microbiome might have bioactivity and contribute to the occurrence of OAB syndrome [70]. Therefore, the composition of the urinary microbiome has its protective or pathogenic roles in the health of the urothelium and modulating bladder function [71].

Several potential mechanisms were provided to explain how the urinary microbiome could behave in the pathogenesis of OAB syndrome [71]. (Figure 2B) First, the bacteria floras may produce neurotransmitters (e.g., ATP) to elicit afferent impulses and then induce bladder contraction [72]. Second, commensal bacteria (e.g., Lactobacillus) could reduce the load of virulent pathogens [73]. Third, commensal bacteria could break down other noxious compounds in urine [71]. Fourth, the microbiome could regulate and maintain urothelial functions by reinforcing the epithelial junction and activating immunological defenses as mechanical barriers [71]. Fifth, commensal bacteria can be crucial for the proper development of the urinary tract, including the urothelium and peripheral nervous system [71].

Clinically, researchers have reported the differences in urinary microbiome between women with urgent urinary incontinence and controls. Pearce et al. found an increase in Gardnerella and a decrease in Lactobacillus in urinary microbiome were associated with female urgent urinary incontinence [73]. By using network analysis to study the microbiota between the vagina and bladder, Nardos et al. pointed out a loss of diversity in urinary microbiota, which means the bladder bacterial genera are more likely to overlap with vagina bacterial genera, and this could be associated with female urgent urinary incontinence (43% in incontinent women vs. 29% in controls) [74].

### 8.2. MetS Alters the Microbiome and Might Shape a Subtype of OAB

MetS may originate from a sedentary lifestyle and the Western diet, as the aberrant gut microbiome of hosts might have the influence on the development of MetS [75,76]. The balance between metabolically healthy microbiota and dysbiosis is crucial for human metabolism. A high-fat and low-fiber diet may induce intestine dysbiosis resulting in increased gut permeability, and thereby translocation of bacterial components into the circulation [75,77]. Bacterial translocation may lead to inflammation in several tissues (e.g., beta-cell, adipose tissue, and liver), and consequently, loss of function. Hyperglycemia and adiposopathy can further induce a proinflammatory response of the immune system through the so-called “immunometabolism” [76].

The framework containing the microbiota in human diseases is not only limited to the metabolic diseases. Nowadays, the concept of the bladder–gut–brain axis provides a more comprehensive role of the microbiota in functional urological disorders [78]. Bidirectional communication within this global axis modulates lower urinary tract function through a defensive response of central sensitization, which means that OAB could be induced by either physical or psychological threats. This bladder–gut–brain axis also indicates the implications of pelvic organ cross-talk and the significance for treating coexisting pelvic disorders together [79,80]. We proposed that MetS can alter the diversity of the microbiota in the urogenital organ and subsequently form a unique subclassification of OAB, namely, MetS-dysbiosis-associated OAB.

## 9. Conclusions

The etiology of OAB contains various phenotypes. In this article, we highlighted the overlapping contributors between MetS and OAB syndrome, including autonomic nervous system dysfunction, chronic ischemia, chronic low-grade proinflammation, dysregulation of nutrient-sensing pathways, and dysbiosis. Secondary to the aforementioned common pathophysiologies between MetS and OAB, the imbalance between the production of pro-oxidants and their elimination through antioxidant system would further damage the bladder function, such as bladder denervation, fibrosis, and bladder afferent oversensitivity [30].

For such subtypes of MetS-associated OAB, nonspecific pharmacotherapies (e.g., antimuscarinics, β-3 agonist, and botulinum toxin injection) might lead to unsatisfying results, although current data remain controversial [6]. For instance, integrated care for MetS-dysbiosis-associated OAB should encompass both the control of MetS and the restoration of the urogenital microbial ecosystem. Meanwhile, the MetS-associated OAB might have potential biomarkers for the diagnosis and therapeutic targeting of both MetS and OAB. However, more collaborative studies for the interaction between MetS and OAB syndrome are required. In this era, high-throughput technologies, such as next-generation sequencing [81] and metabolomics [82], are available to analyze the relationship between MetS and OAB. In addition, OAB must be based on the symptom of urgency and should not be confused with other storage symptoms induced by medical disease [83]. Therefore, machine learning algorithms have potential for differentiating true OAB [84], analyzing brain activity for central sensitization [85], and detecting MetS early [86].

## Figures and Tables

**Figure 1 biomedicines-10-01957-f001:**
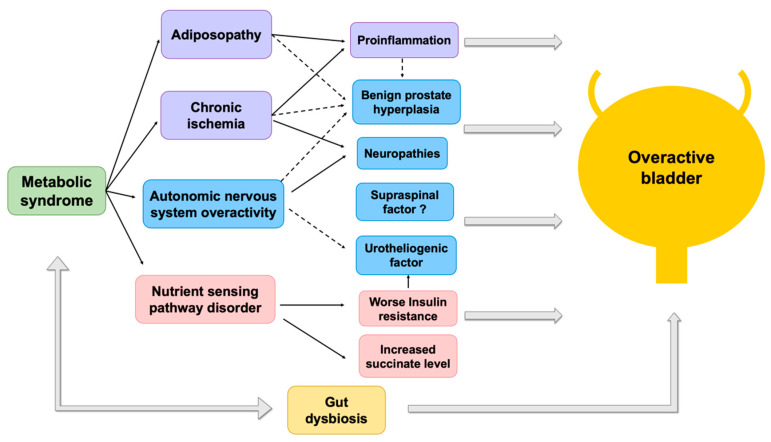
Pathophysiological relationships between metabolic syndrome and overactive bladder. Solid line represents direct effects. Dotted line represents indirect effects.

**Figure 2 biomedicines-10-01957-f002:**
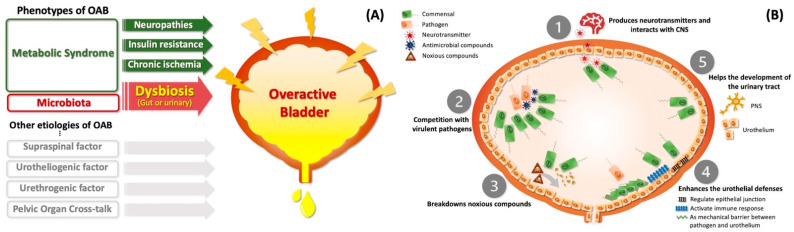
**Potential mechanisms of metabolic syndrome-dysbiosis-associated overactive bladder.** (**A**). Gut or urinary dysbiosis is one of the overlapping mechanisms between metabolic syndrome and overactive bladder. Gut flora can migrate to the urinary bladder via vagina colonization in women [74]. (**B**). Five potential mechanisms of how the urinary microbiome contributes to overactive bladder. (1). Produces neurotransmitters interplaying with the central nervous system. (2) Competes with virulent pathogens. (3) Degrades other urinary noxious compounds. (4) Regulates and maintain urothelial functions by switching on the immune defenses and strengthening the mechanical barrier. (5) Contributes to the development of the urothelium and peripheral nervous system of the urinary tract [71]. **Abbreviation:** ANS, autonomic nervous system. CNS, central nervous system. PNS, peripheral nervous system.

**Table 1 biomedicines-10-01957-t001:** Definitions of metabolic syndrome.

	Key Concept	Criteria	Obesity	Blood Pressure	Dyslipidemia	Hyperglycemia	Others
WHO (1998) [10]	Consensus Definition	Insulin resistance or diabetes, plus two of the other criteria below	Waist/hip ratio: >0.90 in men, >0.85 in women; or BMI >30 kg/m^2^	≥140/90 mmHg	TG 150 mg/dL; HDL-cholesterol <35 mg/dL in men, <39 mg/dL in women	Insulin resistance ^‡^	Microalbuminuria *
EGIR (1999) [11]	Hyperinsulinemia	Hyperinsulinemia, plus two of the other criteria below	Waist circumference: ≥94 cm in men, ≥80 cm in women	≥140/90 mmHg or Rx	TG ≥ 177 mg/dL or HDL-cholesterol <39 mg/dL	Insulin resistance ^‡^	
NCEP:ATP III (2001) [12]		Any three or more of the criteria below	Waist circumference: >102 cm in men, >88 cm in women	≥130/85 mmHg	TG ≥ 150 mg/dL; HDL-cholesterol <40 mg/dL in men, <50 mg/dL in women	Fasting glucose ≥110 mg/dL	
NCEP ATP III (2005 revision) [13]	Central obesity	Any three of the criteria below	Waist circumference: >40 inches in men, >35 inches in women	≥130/85 mmHg or Rx	TG ≥ 150 mg/dL; HDL-cholesterol <40 mg/dL in men, <50 mg/dL in women	Fasting glucose ≥100 mg/dL	
IDF (2005) [14]		Central obesity with ethnicity-specific values ^§^, plus two of the other criteria below	Central obesity with ethnicity-specific values ^§^	≥130/85 mmHg	TG ≥ 150 mg/dL; HDL-cholesterol <40 mg/dL in men, <50 mg/dL in women	Fasting glucose ≥110 mg/dL	
Consensus Definition [15]			Elevated waist circumference (according to country-specific definitions)	≥130/85 mmHg	TG ≥ 150 mg/dL; HDL-cholesterol <40 mg/dL in men, <50 mg/dL in women	Fasting glucose ≥110 mg/dL	

^‡^ Insulin resistance is defined as type 2 diabetes mellitus or impaired fasting glucose (>100 mg/dL) or impaired glucose tolerance. * Urinary albumin excretion of 20 μg/min or albumin-to-creatinine ratio of 30 mg/g. ^§^ To meet the criteria, waist circumference must be: for Europeans, >94 cm in men and >80 cm in women; and for South Asians, Chinese, and Japanese, >90 cm in men and >80 cm in women. For ethnic South and Central Americans, South Asian data are used, and for sub-Saharan Africans and Eastern Mediterranean and Middle East (Arab) populations, European data are used. Rx, pharmacologic treatment.

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
