# Peer review of "Metabolic Syndrome and Overactive Bladder Syndrome May Share Common Pathophysiologies"

_biomedicines, 2022, doi:10.3390/biomedicines10081957_

Round 1

Reviewer 1 Report

Thank you very much to the Editor of Biomedicines for allowing me to review the paper entitled
Metabolic Syndrome and Overactive Bladder Syndrome May Share Common Pathophysiologies
Main observations:
1. The chosen study design is appropriate.
2. The manuscript topic is consistent with the journal content.
3. The manuscript's contribution to the paper's topic can be considered limited.
4. I believe this study would be a candidate for publication in your journal as a review article, with major revisions.
5. Lack of LIMITATION section (at the end of the Discussion section).
Literature is relatively out of date - more than 63% are articles more than five years old - should use more current items.

Minor comments:
Instead of:
For the convenience of scientific investigation, the pathophysiologies of OAB could be classified as myogenic, neurogenic, urotheliogenic, and other specific conditions (e.g., MetS, diabetes, proinflammation, benign prostatic hyperplasia (BPH), affective disorders, urinary microbiota, etc.).
Should be:
For the convenience of scientific investigation, the pathophysiologies of OAB could be classified as myogenic, neurogenic, urotheliogenic, and other specific conditions (e.g., MetS, type 2 diabetes, proinflammation, benign prostatic hyperplasia (BPH), affective disorders, urinary microbiota, etc.).
Instead of:
In Baltimore Longitudinal Study of Aging, researchers reported that each kg/m2 increase in body mass index was associated with a 0.41ml increase in prostate volume.
Should be:

In the Baltimore Longitudinal Study of Aging, researchers reported that each kg/m2 increase in body mass index was associated with a 0.41ml increase in prostate volume.
Instead of:
In an obese mice model, Leiria e al. demonstrated that defective insulin action in bladder mucosa impaired the detrusor relaxation and contributed to bladder overactivity.
Should be:
In an obese mice model, Leiria et al. demonstrated that defective insulin action in bladder mucosa impaired the detrusor relaxation and contributed to bladder overactivity.

Author Response

Responses to the comments by the Reviewers and Editor on manuscript ID: biomedicines-1861816

We thank the Reviewer’s suggestions. We revised the manuscript as the request. Our responses are presented below.

Minor comments:
Instead of:
For the convenience of scientific investigation, the pathophysiologies of OAB could be classified as myogenic, neurogenic, urotheliogenic, and other specific conditions (e.g., MetS, diabetes, proinflammation, benign prostatic hyperplasia (BPH), affective disorders, urinary microbiota, etc.).
Should be:
For the convenience of scientific investigation, the pathophysiologies of OAB could be classified as myogenic, neurogenic, urotheliogenic, and other specific conditions (e.g., MetS, type 2 diabetes, proinflammation, benign prostatic hyperplasia (BPH), affective disorders, urinary microbiota, etc.).

Reply: Authors thank the reviewer. We have revised it.

Instead of:
In Baltimore Longitudinal Study of Aging, researchers reported that each kg/m2 increase in body mass index was associated with a 0.41ml increase in prostate volume.
Should be:

In the Baltimore Longitudinal Study of Aging, researchers reported that each kg/m2 increase in body mass index was associated with a 0.41ml increase in prostate volume.

Reply: Authors thank the reviewer. We have revised it.

Instead of:
In an obese mice model, Leiria e al. demonstrated that defective insulin action in bladder mucosa impaired the detrusor relaxation and contributed to bladder overactivity.
Should be:
In an obese mice model, Leiria et al. demonstrated that defective insulin action in bladder mucosa impaired the detrusor relaxation and contributed to bladder overactivity.

Reply: Authors thank the reviewer. We have revised it.

Reviewer 2 Report

Authors studied important aspects about MetS associated to OAB.  The paper is well organized and structured. I suggest a minor revision.

Strengths:

-Clear presentation of the results;

-Analytical descriptions about the Mets and OAB parameters;

-Study about the overlapping contributors between MetS and OAB syndrom,

- Good graphical models describing the analysed parameters. 

Weaknesses:

-Authors in the conclusion affirm that "more collaborative studies for the interaction between MetS and OAB syndrome are required". Please specify the future approaches about this point.

-Authors should introduce new approaches/methods supporting the analysis such as machine learning algorithms

According to the last point of weaknesses i suggest to improve the state of the art with works about algorithms supporting correlations/classifications/risks analysesm prediction and parameter clustering (in different medical fields), such as:

 DOI: 10.1109/MeMeA49120.2020.9137224
- https://doi.org/10.3390/s22145247
- https://doi.org/10.3390/s22113966

As perpsectives I suggest to introduce machine learning As a possible improvement of the performed analysis.

Author Response

Responses to the comments by the Reviewers and Editor on manuscript ID: biomedicines-1861816

We thank the Reviewer’s suggestions. We revised the manuscript as the request. Our responses are presented below.

Authors studied important aspects about MetS associated to OAB. The paper is well organized and structured. I suggest a minor revision.

-Authors in the conclusion affirm that "more collaborative studies for the interaction between MetS and OAB syndrome are required". Please specify the future approaches about this point.

Reply: We have added sentences at the end of this manuscript, as follows: In this era, high-throughput technologies, such as next-generation sequencing [81] and metabolomics [82] are available to analyze the relationship between MetS and OAB. In addition, OAB must base on the symptom of urgency and should not be confused with other storage symptoms induced by medical disease [83]. Therefore, machine learning algorithms is potential for differential the true OAB [84], analyzing brain activity for central sensitization [85], and early detecting the MetS [86].

References:

  1. Lee, W.C., Tain, Y.L. Maternal Fructose Exposure Programs Metabolic Syndrome-Associated Bladder Overactivity in Young Adult Offspring. Sci Rep. 2016, 6, 34669.
  2. Mitsui, T. Kira, S. Metabolomics approach to male lower urinary tract symptoms: Identification of possible biomarkers and potential targets for new treatments. J Urol 2017, 199, 1312-1318.
  3. Yu, C.C., Hsu, C.C. Medical diseases affecting lower urinary tract function. Urol Sci 2013, 24, 41-45.
  4. Massaro, A., Galiano, A. Telemedicine DSS-AI multi level platform for monoclonal gammopathy assistance. 2020 IEEE International Symposium.
  5. Falco, I.D., Pietro, G.D. A two-step approach for classification in Alzheimer’s disease. Sensors (Basel) 2022, 22, 3966.
  6. Laila, U.E., Mahboob, K. An ensemble approach to predict early-stage diabetes risk using machine learning: An empirical study. Sensors (Basel) 2022, 22, 5247.

-Authors should introduce new approaches/methods supporting the analysis such as machine learning algorithms

Reply: Thanks, the reviewer. Machine learning algorithms is a hot topic. By your suggestion. We have added the possible utility in Machine learning.  

According to the last point of weaknesses I suggest to improve the state of the art with works about algorithms supporting correlations/classifications/risks analysesm prediction and parameter clustering (in different medical fields), such as:

 DOI: 10.1109/MeMeA49120.2020.9137224
- https://doi.org/10.3390/s22145247
- https://doi.org/10.3390/s22113966

As perspectives I suggest to introduce machine learning As a possible improvement of the performed analysis.

Reply: We thanks the reviewer’s suggestion. We have introduced the importance of machine learning in this manuscript.
